# Essential Oil Based PVP-CMC-BC-GG Functional Hydrogel Sachet for ‘Cheese’: Its Shelf Life Confirmed with Anthocyanin (Isolated from Red Cabbage) Bio Stickers

**DOI:** 10.3390/foods9030307

**Published:** 2020-03-09

**Authors:** Smarak Bandyopadhyay, Nabanita Saha, Oyunchimeg Zandraa, Martina Pummerová, Petr Sáha

**Affiliations:** 1Centre of Polymeric Systems, University Institute, Tomas Bata University in Zlin, Tř. T. Bati 5678, Zlin 76001, Czech Republic; bandyopadhyay@utb.cz (S.B.); zandraa@utb.cz (O.Z.); pummerova@utb.cz (M.P.); saha@utb.cz (P.S.); 2Polymer Centre, Faculty of Technology, Tomas Bata University in Zlin, Vavrečkova 275, Zlin 76001, Czech Republic

**Keywords:** anthocyanin, essential oil, PVP-CMC-BC-GG hydrogel film, Gouda cheese, red cabbage extract, food quality, shelf life, intelligent, active, bioactive packaging

## Abstract

‘Gouda cheese’ is one of the most popular varieties of cheese eaten worldwide. The preservation problem of gouda arises due to microbial contamination and infestation. Therefore, essential oil (EO) based PVP-CMC-BC-GG hydrogel film was prepared to solve the problem and to extend the shelf-life of ‘Gouda cheese’. Anthocyanin (isolated from red cabbage) based pH stickers are integrated into the packaging system to recognize the spoilage of ‘cheese’. EOs (clove and/or cinnamon) are added to PVP-CMC-BC-GG hydrogel film to improve its antimicrobial, physical, mechanical, and thermal properties as well as shelf-life of cheese. The films are assessed based on their physical, structural, and functional properties, real-time assessment on cheese, and biodegradability. The results revealed that although the addition of oils to the PVP-CMC-BC-GG hydrogel films enhanced its mechanical, hydrophobic, and antimicrobial properties, the biodegradability of PVP-CMC-BC-GG films declined with the addition of EOs. The thermal properties remained the same irrespective of the addition of EOs. The shelf life of cheese was extended for more than 10–12 days, inside the PVP-CMC-BC-GG hydrogel sachet compared to the conventional PE packaging system. Hence the use of the PVP-CMC-BC-GG sachet (containing EO or without EO) is recommended for cheese packaging along with the use of PVP-CMC-BC-GG anthocyanin bio stickers for monitoring the quality of cheese.

## 1. Introduction

In the food packaging segment, biodegradable materials are in the research and pilot-scale usage for the quick answer they give to the waste management and pollution issues [1]. Polylactic acid, cellulose, starch, and chitosan are the most widely used promising and alternative packaging materials. The justification for using polyvinylpyrrolidone-carboxymethyl cellulose-bacterial cellulose-guar gum (PVP-CMC-BC-GG) based hydrogel films as an alternative food packaging material for berries has been established in our previous study [2]. The current article discusses the preservation of ‘cheese’ using the similar hydrogel composition (i.e., PVP-CMC-BC-GG) with and without EO, (EO used as a food preservative). This study also involves PVP-CMC-BC-GG based pH sticker based on anthocyanin as an indicator of food spoilage.

The safest food preserving agents comes from the natural sources of plants. The EO/s extracted from different plant parts, apart from extending the shelf life of the food [3,4] also, enhance the organoleptic acceptability of the packaged food among customers [5]. Moreover, EOs are being classified safe under GRAS (generally recognized as safe) and QPS (qualified presumption of safety) in the EU and USA respectively [5,6]. Among all the EOs, clove and cinnamon extract oils with eugenol and cinnamaldehyde as the two main active compounds [7] are used from time immemorial for their antimicrobial properties. The shelf life quality of the cheese wrapped with hydrogels can be monitored by the change in pH inside the packaging system, the pH change is directly linked with the food quality change [8]. The anthocyanin from red cabbage (RC) can be used for monitoring the pH for its significant color range for a wide range of pH [9]. The RC extract (RCE) is red at acidic pH and blue/green at alkaline or basic pH (Figure 1).

Besides, the increase in the world market for cheese has made it a good platform for the application of the innovative packages [10] discussed above. ‘Cheese’ is selected in this study because of its popularity in daily life and its nutritional value comprising of fat, vitamins, and inorganic salts. However, cheese is susceptible to the spoilage by pathogenic or nonpathogenic microbes during storage [11,12] resulting in off-flavors [10]. Enhancement of cheese shelf life by using alginate-zein films with natamycin [13] and agar-based hydrogel with silver nanoparticles were reported earlier [14] but using of EO based hydrogel for improvement of the cheese shelf life and endorsing its quality and safety has not yet been reported as per the best of our knowledge. The use of EOs and anthocyanins based pH sticker for improvement and monitoring the cheese is a good approach unless these are added directly to the cheese, which may deteriorate the sensory qualities and make them unfit for consumer acceptance. So, the best option is to cover the cheese with intelligent and active films for product enhancement. Earlier many authors have applied different EOs to cheese directly, but none applied clove and cinnamon at the same time [15]. Therefore, to observe the effect of clove and cinnamon EOs in the hydrogel film and its implementation to evaluate the shelf life of cheese has been reported for the first time.

This study aims to see the effect of EOs, added to the hydrogel films, on the shelf life of cheese packed with the hydrogels. Anthocyanin will be used to detect the spoilage of cheese by monitoring the pH changes in the food. Moreover, in this study focus has also been given on the assessment of the antimicrobial, physical, structural, mechanical, and thermal properties of the PVP-CMC-BC-GG based hydrogel film with and without EOs. Furthermore, the biodegradation analysis of films after the incorporation of EOs has been rarely reported, so to bring some information on this aspect, biodegradability study of PVP-CMC-BC-GG hydrogel films with and without EOs are also performed. The information from this study could be useful for designing future innovative packaging system for ‘cheese’ or similar products.

## 2. Materials and Methods

### 2.1. Materials

The carboxymethyl cellulose (CMC) was purchased from Sinopharm Chemical Reagents Co. Ltd. Polyvinyl pyrrolidone (PVP) and guar gum (GG) were procured from Sigma Aldrich, Prague, CZ. While polyethylene glycol (PEG) and agar were supplied from Fluka, Prague, CZ. Other chemicals like glycerine (Gly) and sodium hydroxide (NaOH) were obtained from Lach-Ner, Brno, CZ, and Penta sro, Prague, CZ respectively. The cinnamon bark and clove bud essential oils were prepared by steam distillation method and supplied by M+H Míča a Harašta s.r.o., Břeclav, CZ. The clove and cinnamon essential oils (EOs) had a specific gravity, optical rotation (589 nm), and refractive index all at 20 °C, which was 1.11 gm/cc, −1.3, and 1.537 and 1.054, −0.25, and 1.609 respectively. All the microorganisms used in this study were bought from the Czech Collection of Microorganisms (CCM), Brno, CZ and maintained in the microbiology department of our university institute. The culture medium tryptone soya agar (TSA), Hinton Mueller Agar (HMA), Sabouraud Dextrose Maltose Agar (SDMA), Mueller Hinton Agar, and 2% Glucose with Methylene blue (HMA, 2% Glu. MB) were bought from HiMedia Laboratories GmbH, Einhausen, Germany. The thin-sliced ‘Gouda cheese’ and red cabbage (RC: *Brassica oleraceae*) were bought from the local supermarket in Zlin. 

BC was produced in our laboratory from a modified apple juice medium (m-AJM) following the method described in Bandyopadhyay et al. [16]. The composition of m-AJM is 0.8 g ammonium sulfate, 0.2 g dipotassium phosphate, 2 g sucrose, and 0.5 mL acetic acid added to 100 mL apple juice. The pH of the m-AJM was adjusted to 5.5 using 0.5N NaOH.

### 2.2. Preparation of Essential Oil-Based PVP-CMC-BC-GG Films

The films were prepared by the casting method in polyvinyl chloride trays. The composition of PVP-CMC-BC-GG film (1 g PVP, 0.5 g CMC, 0.35 g GG, 0.15 g BC, 1.5 g PEG, 3 g Agar, and 1.5 mL Gly added to 150 mL water) was the same as reported in our previous article [2] and was considered as a control/reference sample. The EOs (clove and cinnamon) were added to the polymer mixture at the rate of 2% of the total volume before heat polymerizing. The two EOs were added separately as well as in combination at the rate of 2% in each PVP-CMC-BC-GG polymer solution. The PVP-CMC-BC-GG film with 2% clove EO is referred to as ‘CLEO’ and the PVP-CMC-BC-GG film with 2% cinnamon EO is mentioned as ‘CiEO’ in the later texts. While a synergistic film of the two oils was prepared by adding 1% of each oil to the PVP-CMC-BC-GG polymer solution, stated as ‘CLiEO’. The films were air-dried until 3% ± 0.3% moisture for 24 h at RT (23 °C) and later peeled out using tweezers. The films were stored in a desiccator over silica beads at RT (24 °C) until tested. All the essential oil-based PVP-CMC-BC-GG dry films revealed the properties of hydrogel film (as shown in Figure 2b–e), moreover, the achieved EO based hydrogel films had 90% ± 5% water (before drying) and could absorb 600% ± 50% water (after drying). The films were dried at 60 °C, 20% ± 5% RH in a climate chamber HCP105 (memmert, Germany) until equilibrium for water content measurement.

Gas chromatography and mass spectroscopy of the EOs were performed on an Agilent 7890B (Santa Clara, CA, USA) gas chromatograph with a Q-TOF 7200 mass detector to confirm the presence of an individual component in the EOs. The EOs were dissolved in n-hexane to obtain a final concentration of 1 µL/mL. Two coupled HP-5ms columns (30 m × 0.25 mm × 0.25 µm, Agilent) were used. Helium was the carrier gas with a column flow rate of 1.0 and 1.2 mL/min. The injection volume was 1 µL and the split ratio was 1:200. A temperature program of 100 °C for 3 min followed by 20 °C/min to 250 °C for 5 min was used for the analysis (total time 15.5 min). The mass detector conditions were: EI with ionization voltage 70 eV, Aux heater 280 °C, ion source 230 °C, and mass range 35–350 amu. The spectra were compared with the NIST library.

### 2.3. Preparation of Bio Stickers (Acting as a pH Indicator)

Anthocyanin (isolated from RC: *Brassica oleraceae*) bio stickers were prepared to perform as a ‘pH indicator’ (Figure 1) and to confirm the change of pH in cheese with time. Extraction of anthocyanin from RC was done by little modification of method mentioned by Fuleki and Francis [17]. One hundred and fifty grams of chopped RC was blended with 100 mL of 70% ethanol in a Waring blender at full speed for 90 s. Later the blender is washed with more 50 mL ethanol and was added to the blend. The mixture was kept overnight in a refrigerator at 4 °C for complete extraction of the anthocyanin into the solvent. Thereafter, the mixture was taken out and vacuum filtered using a Buchner funnel and Whatman No1 filter paper. More 100 mL of ethanol was used to wash the residue in the filter paper and added to the 200 mL extracted volume for a final pH of 6.76 at 24.7 °C. The filtered extract was stored in dark for 2 h and a small amount was diluted with ethanol to get the O.D values at 535 nm using Cary 300 UV*-*visible spectrophotometer (Agilent, Santa Clara, CA, USA*).* Now the total anthocyanin content (TAC) in the extract was estimated at 7.11 ± 0.41 mg/100 g using the formulae [17,18]:TAC (mg/100g) = (O.D × DV × VF × TEV)/ (SW × 51.56)
where: OD = Optical density or absorbance reading at 535 nm; DV = Diluted volume for the OD measurements, in our case it is 100 mL; VF = Volume factor is the ratio between the diluted volume and the sample volume, in our case, we took 2%, 5%, and 10% diluted solution so VF would be 100/2, 100/5, and 100/10 respectively; TEV = Total extracted volume, in our case it was 300 mL; SW = Sample weight or weight of RC taken for extraction, in our case it was 150 g; 51.56 = E value or molar extinction coefficient for red cabbage [18].

The RCE was immobilized on the hydrogel films by a modified absorption method reported earlier [19]. The bio stickers were dried and kept in dark on a Petri dish covered with aluminum foil, for overnight before use. The films were cut in a circle with a diameter of 40 mm and three circles are soaked in a 90 mm diameter Petri dish with 5 mL RCE. The Petri dish was kept in a BenchRocker™ 3D shaker (Benchmark, Sayreville, NJ, USA) at 100 rpm. After 30 min they were shifted to vacuum oven for 2 h at 24–27 °C.

### 2.4. Antimicrobial Activity

The antibacterial and antifungal effects of the PVP-CMC-BC-GG films (with and without EOs) were investigated against six common pathogenic and spoilage microbes, usually present in dairy products. Kirby-Bauer disc diffusion method was used for two Gram-negative bacteria (*Bacillus cereus* CCM 7934 and *Staphylococcus aureus* CCM 4516), two Gram–positive bacteria (*E. coli* CCM 4517 and *Klebsiella pneumoniae* CCM 4415) and two fungi (*Aspergillus brasiliensis* CCM 8222 and *Candida albicans* CCM 8215). All the bacteria and fungi were pre-cultured in TSA for 24 h at 35 °C and SDMA for 48 h at 35 °C respectively. The films were cut in circular discs with 6 mm diameter and placed on the pre-inoculated cultures (diluted to 0.5 McFarland turbidity standard) in HMA medium (bacteria) and HMA, 2% Glu. MB medium (fungi). These were then incubated at 35 °C for 24 h. The inhibition zones were calculated using Scan^®^ 500 automatic colony counters (interscience, FR).

### 2.5. Migration Test

The release of the antimicrobial EOs from the films was studied by the UV-Vis spectroscopy method [20]. Since the EOs are correlated to the antimicrobial properties by the films, it is important to measure the release rate of the EOs from the films to the packaging environment. To interact with broader food contact, the EOs are presumed to get a release by water uptake and swelling of the films [21]. The slow and stable release of EO from the film can be correlated to better antimicrobial properties in real-time [21]. All the oil-based films are cut (30 mm × 10 mm) and immersed in 5 mL distilled water kept in BenchRocker™ 3D shaker (Benchmark, Sayreville, NJ, USA) at medium speed for 24 h. The first O.D value (289 nm) of the water after immersion of the films was recorded at 2 h, 24 h, followed by 48 h, and 72 h, changing the water at each interval. The solutions were vortexed for 25 s before O.D readings. It is hypothesized that the O.D values will slowly decrease with time as the films will release the oils in the water, but the aim is to know the film with the best balance between retention and release rate (steady release).

### 2.6. Characterization of the EO Based PVP-CMC-BC-GG Based Films

#### 2.6.1. Scanning Electron Microscopy

The cross-sectional and surface morphology was observed by the method described in Smarak Bandyopadhyay et al. (2019). The images were taken at a magnification of 800× and 5 kV for structural morphology while 800× with 5 kV and 10 kV was used for migration test confirmation.

#### 2.6.2. FTIR Investigations

The structural Fourier transfer infrared (FTIR) spectroscopy was done in an attenuated total reflectance (ATR) mode (Nicolet iS5 by Thermo Scientific, Waltham, MA, USA) following the exact method reported earlier [16].

#### 2.6.3. XRD Analysis

X-ray diffraction technique was applied to know the crystalline structure of the films using a MiniFlex™ 600 X-ray diffractometer (Rigaku, Tokyo, Japan). The method is the same as reported earlier [16].

#### 2.6.4. Color Assay

Color is an important parameter to quantify the visual or optical changes occurring in the PVP-CMC-BC-GG film with the addition of EOs or reacting with other substances. The color assay of the films was done using Lovibond RT 850I (Tintometer^®^, Amesbury, UK) and analyzed by Oncolor^™^ application. ISO standards were followed for all the measurements. Calculating the color parameters for the oil-based films, PVP-CMC-BC-GG film was used as a standard, while PVP-CMC-BC-GG film soaked in RCE was used as a standard for measuring the color differences of the bio stickers with time.

#### 2.6.5. Mechanical Properties

Mechanical analysis is an important part of all packaging material to test its ability to withstand shear and stress. The data are thus represented as the effect of EO on the mechanical properties of PVP-CMC-BC-GG films. The tensile properties of the films were performed in Testometric MT350-5CT (Labomachine, Rochdale, CZ) and data analyzed by winTest™ analysis following ISO standards. All the films were cut in the dimension of 50 mm × 10 mm and the cross-head speed was maintained at 10 mm/min with a static load of 10 kgf was used. The usual thickness of the films was 0.125 mm ± 0.014 and all the data represent an average of five replications.

#### 2.6.6. Oxygen Permeability

The oxygen permeability (OP) is an important parameter in food preservation and was measured following ASTM D1434 standards using PERME™ VAC V1 (Labthink Instruments Co., Ltd., Jinan, China) mentioned in our earlier work [2]. The temperature and RH during the experiment was 26 °C and 32% respectively.

#### 2.6.7. Contact Angle Measurement

The rectangular pieces of films (2 cm × 5 cm) were measured for the contact angle using the See System (Advex Instruments s.r.o, Brno, CZ). This valuation is important for assessing the extent of hydrophobicity of any material and was done exactly as reported in our previous work [2].

#### 2.6.8. Thermogravimetric Analysis (TGA)

Along with other instrumental quantification, TGA is important to judge the thermal stability of the packaging materials and its variation with added components. The average weight of the samples was 6.46 ± 0.28 mg and the protocol followed was according to our previous work [16].

### 2.7. Proof-of-Concept Application

The edges of the films were heat sealed for 7 s with FRN-600 Plastic Film Sealer (HotAir^®^, Ostrava, CZ) to sachet dimension 120 mm × 120 mm and the fresh Gouda cheese (average 18.5 g) slice was kept inside each sachet/packet. The bio-sticker was placed inside each sachet keeping in direct contact with the cheese slice. All the test packets (PVP-CMC-BC-GG hydrogel sachet with and without EO) and a conventional polyethene (PE thickness 0.03 mm) packet with cheese, as shown in Figure 2a, were kept at 4° C for 1 month. The side of the film in touch with the tray and with more EO was faced towards the cheese while packaging. The biosticker was checked for color change at regular intervals. The spoilage indication on the cheese was also monitored in parallel by visual appearance, odor, and structural integrity. The weight loss percentage (W %) from each packaging system was calculated by the formula:W% = [(W_i_ − W_d_)/W_i_] × 100
where W_i_ is the weight of the package on the first day and W_d_ is the weight of the package on the day of reading.

### 2.8. Biodegradation Studies

The soil burial study of the films was done following ASTM G160-12 to evaluate the biodegradability of the EO based hydrogel films. Rectangular pieces of films (50 mm × 50 mm) were cut and kept horizontally in approximately 215 g of soil in a 250 mL beaker. The soil had a pH of 6.7–7.0 and 20% moisture. The experiment is conducted in a humidity chamber with humidity and temperature maintained at 85–90% surrounding humidity and 30 °C respectively. Every fifteen days the films were removed from the compost and washed with distilled water, dried to constant weight before weighing.

The degradation percentage of the films in the soil over 60 days was calculated using the formula [22]:D % = (∆W/W) × 100
where “D is the degradation percentage, ∆W is the change in weight of the films at different buried time, and W is the initial weight of the films” [2].

### 2.9. Experimental Design and Statistical Evaluation

A single-factor split-plot—CRD (completely randomized design)—was done for each film on all the microorganisms, with five replications for each film on each microbe. The data obtained were analyzed by one-way ANOVA using SPSS Ver. 21.0 (IBM, Armonk, NY, USA) software package for Windows. The postho*c* pairwise comparison of the mean was performed with a general linear model test followed by the Tukey HSD method with an adjustment of covariates. The Shapiro–Wilk test was initially followed to determine a normal distribution for each variable before statistical testing, later logarithmic transformation was performed on the skewed variables. The mean and standard error (of the mean) was calculated for each film for individual microorganism. The mean values are presented in untransformed forms for descriptive purposes. A two-tailed value of *p* < 0.05 was considered as statistically significant.

All the other results were the representation of standard deviation showed by the ± symbol after mean values from at least three replications.

## 3. Results and Discussions

### 3.1. Structural and Morphological Analysis

Figure 2b–e depicts the presence of porous structure in the hydrogel, which allows the exchange of gases through the films. The pictures show that the addition of oils had very little effect on the porosity of the films but later it was also discussed that the addition of oils had significant effects on the oxygen permeability. The oils were attached in numerous small vesicles on the surface of the films (Figure 3). This is maybe due to the movement of the oils from the polymer mixture to the film surface while drying and eventually settling on the surface [23]. The FTIR images in Figure 2f show the presence of peaks for cinnamaldehyde at 1656–1673 cm^−1^ [24] and eugenol at 1511 cm^−1^ [25]. The changes in the absorbance pattern of the CiEO and CLiEO from 1332 to 914 cm^−1^ wavenumbers, suggests the formation of new intermolecular bonding between PVP-CMC-BC-GG and cinnamon oil. The XRD graph in Figure 2g is in line with our previous reports [2] and shows the semi-crystalline nature of all the films irrespective of the addition of oils. The peaks found from the XRD are because of the polymer blend. It has been observed as a signature peak for PVP-CMC-BC-GG at 18.9° in our previous study also [2], maybe the crystalline peak for BC has shifted from 17.22° after mixing with other copolymers. The peak also suggests good mixing of BC with the other components. The peak for PVP-CMC blend is reported at 23.09° [2].

### 3.2. Antimicrobial Properties

Cinnamaldehyde and eugenol are the main antimicrobial components present in the cinnamon and clove EOs respectively [26]. The presence of these two essential components cinnamaldehyde and eugenol in our EOs and films were confirmed from the GC-MS (Figure 4) and FTIR analysis (Figure 2f) correspondingly. In this study, all the chosen microorganisms are typical foodborne pathogens and food spoiling agents [27,28,29]. Although, all the EO based films show antimicrobial properties, which is similar to other results reported earlier [7,30,31] the results from our study (Table 1) showing CiEO better inhibition properties than clove oil also lies in agreement with results reported previously [20]. CiEO has shown better inhibition properties at a lower concentration in our study in comparison to 50% (*w*/*w*) cinnamon EO in PLA against gram –ve and gram +ve bacteria [32] and 20% *v*/*v* cinnamon EO against *E. coli*, *B. cereus,* and *S. aureus* [33].

It was also suggested by Villegas et al. [24] that the synergistic effect among the different components present in cinnamon EO (Figure 4) have made them stand apart with better inhibition properties from other EOs. Table 1 and Figure 5 show that the inhibition effects of EO based films were more promising against Gram –ve bacteria, which is in agreement with previous authors [7]. Even, in our study, the antifungal activity of the EO based films were better than antibacterial activities, which supports the earlier findings with 10% EO (*w*/*w*) in chitosan films [30].

As evident from Table 1, CLiEO established a synergistic effect between clove and cinnamon EO against Gram –ve *S. aureus*, Gram +ve *Klebsiella,* and black mold *Aspergillus*. The results in Table 1 show statistically similar results between CiEO and CLiEO against *B. cereus*, *E. coli,* and *Candida*. CLiEO is most effective and unique for its results against *Aspergillus* sp.

### 3.3. Migration Rate of EOs

Since water is a universal solvent we tried our migration tests with water as reported successfully by earlier reports [20,34]. Moreover, it is also proved that the water can migrate the oils from the films. Figure 3 illustrates the release of cinnamon and clove EOs in the solvent for 72 h. The release rate of clove EO from the film was higher than the release rate of cinnamon EO and the mixture of both EOs (Figure 3g). The migration rate of the EOs depends on the strength of their bonding with the hydrogel along with the affinity and solubility of the EOs in the solvent [35]. The diffusion rate of the EOs in the water is an important aspect while judging them from their endpoint of use. Although all the films released 80% of the EOs in the first 24 h, it is important to note that the experimental condition was far more aggressive than the real-life situation. The films have shown swollen properties after the release, similar to Souza et al. [20], but the films did not release all the EOs after 2 h, as reported in their study. This may be due to the stronger bonds of the EO with PVP-CMC-BC-GG. After 72 h all the films released all their EOs, which was also confirmed by the SEM images (Figure 5a–f). CLiEO with the slowest release rate could be further evaluated on other solvent mixtures with different EO concentration at varying temperatures. The reason for this variation in the release rate from the same hydrogel matrix is very complex and can only be attributed to the contribution of each component of the oil in the formation of bonds with the matrix. It is also mentioned by earlier authors [35] that the release rate of the EOs in the real-time packaging environment will depend upon the materials packed, the composition of the food, temperature, humidity, and duration of contact with the food.

### 3.4. Color Assay

Although any material’s color properties are relative in nature, they are also an important part in evaluating any packaging material. The material stability, component adulteration, or browning effects are quickly assessed with the color aspects [36]. It is mentioned in Table 2, all the antimicrobial films have significant (∆E ≥ 5) color difference from the control film. All the samples are proved to have their color profiles that are different from the control film, thus we can compare the films on hue, chroma, brightness, lightness, and haze. Hue is the color, which each individual perceives through their stimuli. So, if one says hue is synonymous with color, and if ∆H somehow represents the color differences from the standard, it will not be wrong. Throughout our analysis, the ∆H follows the values from ∆E and reflects a normal color variance. As described earlier in our previous study [37] that chroma (C) means the purity of color and brightness (Br) features lightness to brightness; thus in comparison to the standard PVP-CMC-BC-GG film, CLEO has the least color purity and is lighter than others. CLEO is also more yellowish with a whiter color component. CLEO will be nearly white with a small reddish-yellow component on the CIE L* A* B* Color Space, CiEO will be little yellow with some greenish component and CLiEO will be less yellow with a less greenish tinge from CiEO.

The ∆L and L* values will give information on the difference in lightness between two samples, supporting the obtainability of the lighter component in CLEO in standalone or in comparison to the standard. Now, haze is a more optical property rather than color property but an important factor in selecting packages. Consider having food packaging films with low haze, more visibility, and transitiveness can be more convenient for the consumers to check the product quality inside. Thus CLiEO with least haze percentage can be more appealing to the end-users. The raised percentage of haze in the other two films may be due to the surface roughness or irregularities in the crystalline structure [38].

### 3.5. Physical Qualities

Table 3 shows the oxygen transmission rate (OTR) of the EO based films. The OP or OTR analyses were not frequently reported due to the increase in their values with the addition of EOs [6], although it is important in the film formulation to prevent food’s degradation by lipid oxidation. The same trend was also detected in CLEO and CiEO films when the percentage of EO was high, but a reverse trend is observed in the synergistic film CLiEO. The decrease of OTR of the films with the addition of less amount of EOs is in agreement with the works of other authors [39,40]. The reason of the decline in the OTR values is mentioned by the same authors and could be due to the homogeneous distribution of the little proportion of EOs in CLiEO [39,40] or may be because of the obstruction created by the water molecules, which replaced the void created by the EOs when they settled to the lower layers. Increase in crystallinity may also be a cause for the decrease in OTR but in our study, this has negligible effect as confirmed from Figure 2g.

The contact angle is a direct evaluation of the material’s hydrophobicity. The two parameters are directly proportional indicating the wettability of the biofilms and its protecting capacity of the packed food. The difference in the contact angle between the upper and lower side of the films, as evident from Table 3 that the oils have moved to the lower side of the films while dying. There was an increase in the contact angle in the lower side of the PVP-CMC-BC-GG with the addition of EOs. This is due to the hydrophobicity of the oils as and often described by the authors [6,39]. In contrary, the upper side of the PVP-CMC-BC-GG film has lost its hydrophobicity with the addition of EOs and their drying process. This may be due to the pushing up of the hydrophilic components (CMC, PEG, etc.) in the hydrogel composition by the oils as they make their path downwards.

### 3.6. Thermo-Mechanical Evaluations

Figure 6 shows the two steps thermal degradation of all the films at 250 ± 2.6 °C (may be due to the degradation of EOs, which also coincides with the BC degradation temperature) and 420 ± 2 °C (may be due to the degradation of other components in the films). The decomposition process finished at around 500 °C with almost 80% weight loss for all the films. The mechanical behavior of the films is reported in Table 4 and Figure 6e. In our study, we observed an increase in the elasticity and tensile strength along with a decrease in the elongation at break, with the addition of EOs to the PVP-CMC-BC-GG film, which is similar to earlier reports [6,39]. The reason for the increase in the tensile properties may be due to the strengthening effect created by the rearrangements of the polymer network with the addition of the EOs or may be due to the development of a strong crosslinking bond between EO and the polymer matrix. The formation of the same strong crosslinking structure may have hindered the elongation properties [6]. Other suggested reasons for the decrease in the ε may be due to the increase in the pore size of the films with the addition of EOs that eventually created probable rupture points. The tensile and elongation properties among the EO based films may vary due to their different degree of hydrophilicity, as the hydrophilicity decreases, modulus and tensile strength decreases, whereas with the increase in hydrophilicity elongation at break increases.

### 3.7. Active and Intelligent Packaging

The EO based antimicrobial films showed promising results in vitro (Figure 5). The main action mechanism of the EOs depends on their hydrophobicity and making the microbe cell membranes more permeable. The major target to disturb the permeability lies in the amount of cellular phospholipid content. Importantly, the phospholipid in the microbial membrane has a higher level of unsaturation at 4−7 °C (food storage condition) than at 35 °C (incubation condition during in vitro antimicrobial analysis) to maintain fluidity and cellular function, thereby attracting more EOs to increase the cellular permeability and eventual death [31]. Thus the films showing inhibition zones in-situ had the potential to show better biocidal properties at low temperatures in the real application when packed with cheese or other products at 4 °C. Although no microbial spoilage of the cheese was noticed in any of the hydrogel-based packings at 4 °C, significant degradation of the cheese was observed in PE packages (Figure 7a). The cheese packed in the hydrogels had only enough moisture to keep the cheese surface moistened while releasing the excess moisture through the pores. This kept the cheese packed in the hydrogel films, fresh and soft over a longer duration of time. While the cheese packed in the PE lost its structural integrity and freshness over the same period. Although there is much emphasis given on the regulations regarding cheese manufacturing the defined acceptable cheese quality from the shelf is not widely reported and depends widely on sensory evaluation [41]. According to a report in sciencealert, the more soft and moist a cheese the easier it gets spoiled [42]. Usually, the Gouda cheese remains fresh for 30 days as evaluated and reported by Saravani et al. [43]. The spoilage of the cheese in the PE (commercial packaging material for soft Gouda slices as we have bought from the market) was precisely marked with the color change in the PVP-CMC-BC-GG based pH indicator films.

The spoilage of food including cheese can be detected using pH variation apart from foul odor and structural changes [44]. The total anthocyanin content in our pH indicator lies between the values reported by the previous journal [45]. The pH films initially turn slightly pinkish when placed on the cheese for the first time due to the interaction with the lactic acid and other organic acids present in the cheese. In Figure 1 it is evident that the pH indicator turns red in the presence of acid due to conversion of anthocyanin in the pH film to red flavylium form and producing green color in the presence of base due to the formation of carbinol pseudobase [8]. Although the colors are interchangeable with low exposure to the acidic or basic agents [8] we have not tried it under this experiment. The pH indicators turn red eventually with time (Figure 7a) in the PE packaging due to the accumulation of lactic acid, pyruvate, acetate, or other organic acids by the metabolic activities of the microfauna [46]. The lowering of the pH in the PE package may be due to the symbiosis between the yeast and the bacteria [47], in contrast to the hydrogel environment where the pH remains almost the same due to the restriction in the post-processing microbial growth by curbing the water content in the packaging environment. Though a lactic acid bacteria (LAB) starter culture is an inevitable component in cheese making under low temperature, low oxygen, and acidic conditions LAB becomes the chief spoiling agent [28]. The spoilage mechanism by LAB works hand in hand with the yeast, as the end products from bacterial metabolism are utilized by the yeasts like *Candida* and in turn, the yeasts produce amino acids, vitamins, and acidic environment for bacterial proliferation. Ageing or ripening is an important part of the cheese-making protocols [48], with proteolysis by LAB being the pivotal process to give the cheese its texture and color. In soft cheese, if the proteolysis is extended for a longer period the cheese will become very soft, whitish with more liquefaction of the solid portion, and a strong ammonia odor. This extended form of proteolysis was observed in the cheese packed in PE to a point of no return after 20 days (Figure 7a). Similarly, Table 5 describes the different color changes in the pH film by values representing the absorbance by the film, a particular wavelength/s of light from the diffused transmitted light. Every L* a* b* C* and haze values of the individual pH indicators in all the packaging environment differ from each other from the 10th day of observation. This can be explained by two reasons, firstly it may be due to the migration of EO components from the packages to the pH film without causing pH reactions and secondly it may be due to pH reactions within the packages. It is important to note the hue of the pH indicators, which remained constant for all the packages except PE, where the color changes with time were significant and prominent. Thus the colorimetric values prove the efficacy of the pH indicator films for cheese shelf life analysis.

### 3.8. Biodegradation Study

The polysaccharides used in the preparation of the films were degraded by the enzymes produced from the soil microbes, which cleaves the glycosidic linkages in the polymer chains [49] leaving behind other free moving components like PEG, which also degrades slowly [50] and PVP, which as per American Polymer Standards Corporations Polyvinylpyrrolidone Safety Data Sheet [51], may remain in the soil with no reported adverse ecological effects. Figure 7b represents the appearance of the films fragmented with degradation. There were no collectable fragments for control and EO films (CLEO and CiEO) after 30 days and 45 days respectively. Some fragments were still collectable for CLiEO, but the entire film has degraded to more than 95% by that time. The films swelled and softened under moisture condition in the soil causing disintegration and degradation of the films. The appearance of black spots in the films after washing is due to the penetration of the soil content and diffusion of soil moisture through the films, suggesting degradation in progress. The rate of degradation for all the films is shown in Figure 7c. Once the film structure disintegrates, the EO bound to the films will get released to the soil [52] inhibiting soil microbiota and affecting the rate of degradation by the microbes. The film with oils will have a slower degradation rate than the control film. The oils in the films will inhibit some of the soil microorganisms but it will not completely remove the soil microbiota. Thus still there will be degradation of the EO based films but at a slower rate. This is prominent from Figure 7c. Now the degree of inhibiting the soil microbiota will depend on the inhibiting power of the individual EO, this is also proved by different degradation rate among EO based films. The inhibition of soil microbe by cinnamon EO is more than clove EO [53] thus the degradation rate of CLEO is more than CiEO (Figure 7c). The degradation rate for CLiEO was higher than CLEO and CiEO may be due to the presence of EOs in the films below inhibitory dose for each EO or the absence of a synergistic effect against soil microbiota.

## 4. Conclusions

The addition of EOs to the PVP-CMC-BC-GG hydrogel films has enhanced its in vitro antimicrobial properties, hydrophobicity, and mechanical properties. The slow and steady migration rate of EOs from the CLiEO hydrogel film also suggest its effectivity in microbial control for a wider time. The oxygen barrier properties of the control film (i.e., PVP-CMC-BC-GG) declined with the addition of individual EOs but only enhanced when both the oils are added together in lower amounts, which need to be evaluated further under different conditions. The thermal properties remained unaffected with the addition of EOs to the PVP-CMC-BC-GG hydrogel film. CLiEO stands best among the EO based PVP-CMC-BC-GG films when compared to antimicrobial properties and color values. The freshness assessment of cheese is successfully confirmed by the anthocyanin based pH stickers placed inside the packages/sachets (as shown in Figure 7b). While the cheese in PE showed spoilage and quality deterioration after 18–20 days, the cheese packed in hydrogel films (with and without EO) remained fresh for 30 days under refrigerated condition (4 °C). Moreover, all the PVP-CMC-BC-GG films were 95% ± 5% biodegraded within 60 days under moist soil conditions. Thus the present work recommends the consideration of PVP-CMC-BC-GG hydrogel film (with and without EO) as an active and functional packaging material for cheese preservation and storage, which has successfully enhanced the shelf life of cheese than conventional packaging. In future, the microbial and sensory evaluation of the cheese can also be done to further prove the effectivity of the PVP-CMC-BC-GG hydrogel films.

## Figures and Tables

**Figure 1 foods-09-00307-f001:**
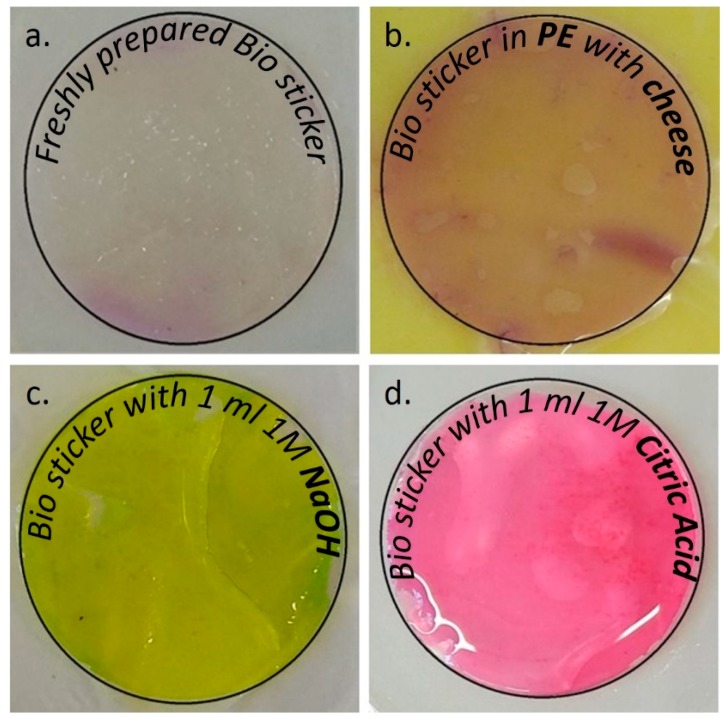
Appearance of the anthocyanin (isolated from red cabbage) based bio sticker: (**a**) freshly prepared, (**b**) in the presence of ‘cheese’, (**c**) in the presence of a base, and (**d**) in the presence of acid.

**Figure 2 foods-09-00307-f002:**
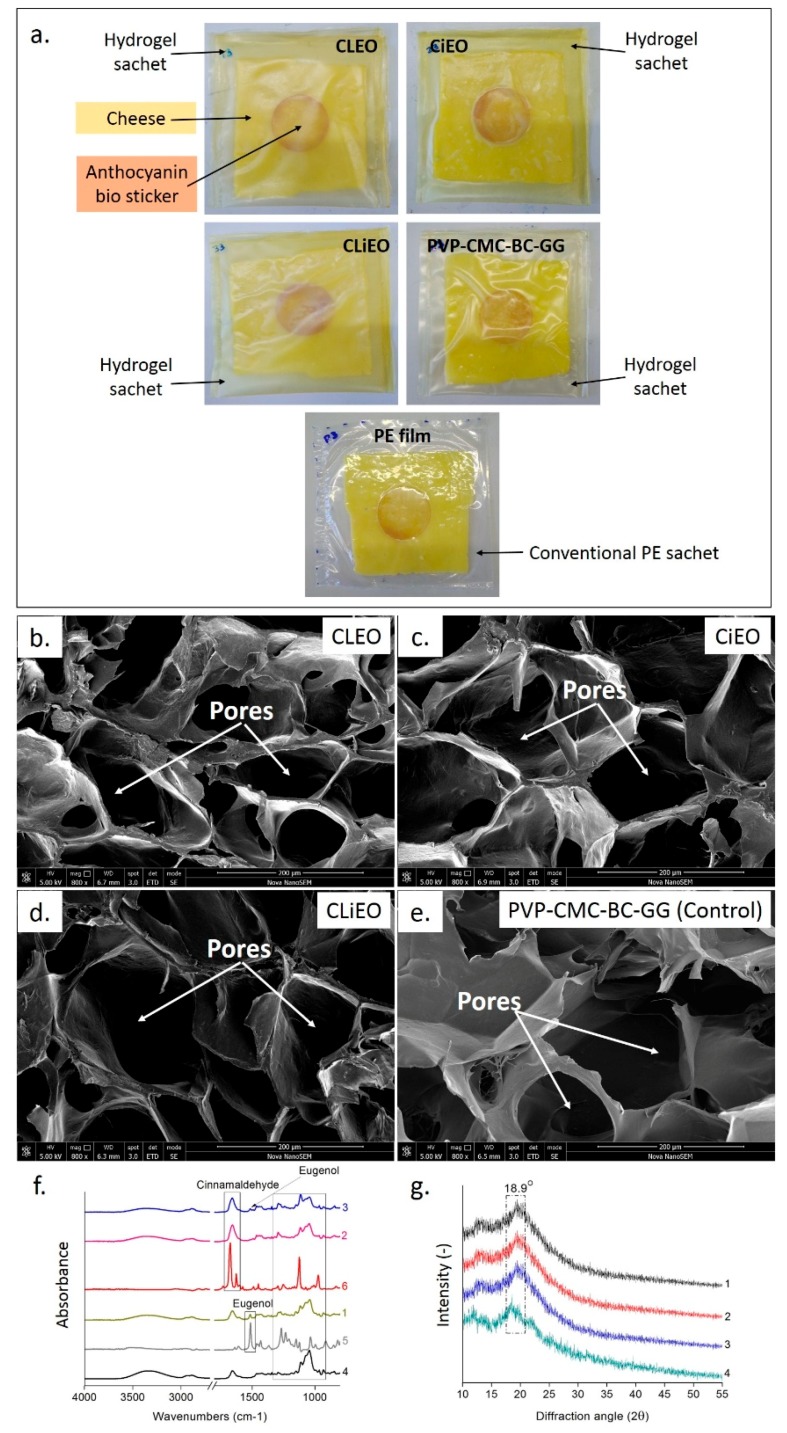
(**a**) Physical appearance of the PVP-CMC-BC-GG hydrogel sachets and conventional PE packet with slice of ‘Gouda cheese’ and anthocyanin bio stickers; (**b**–**e**) structural properties (internal SEM images) of hydrogel film with and without essential oils (EOs) named: CLEO, CiEO, CLiEO, and PVP-CMC-BC-GG; (**f**) FTIR spectra; and (**g**) XRD spectra: *1-*CLEO, *2-*CiEO, *3-*CLiEO films, *4-*PVP-CMC-BC-GG (*Control*) films, *5-*Pure clove oil, and *6-* Pure cinnamon oil.

**Figure 3 foods-09-00307-f003:**
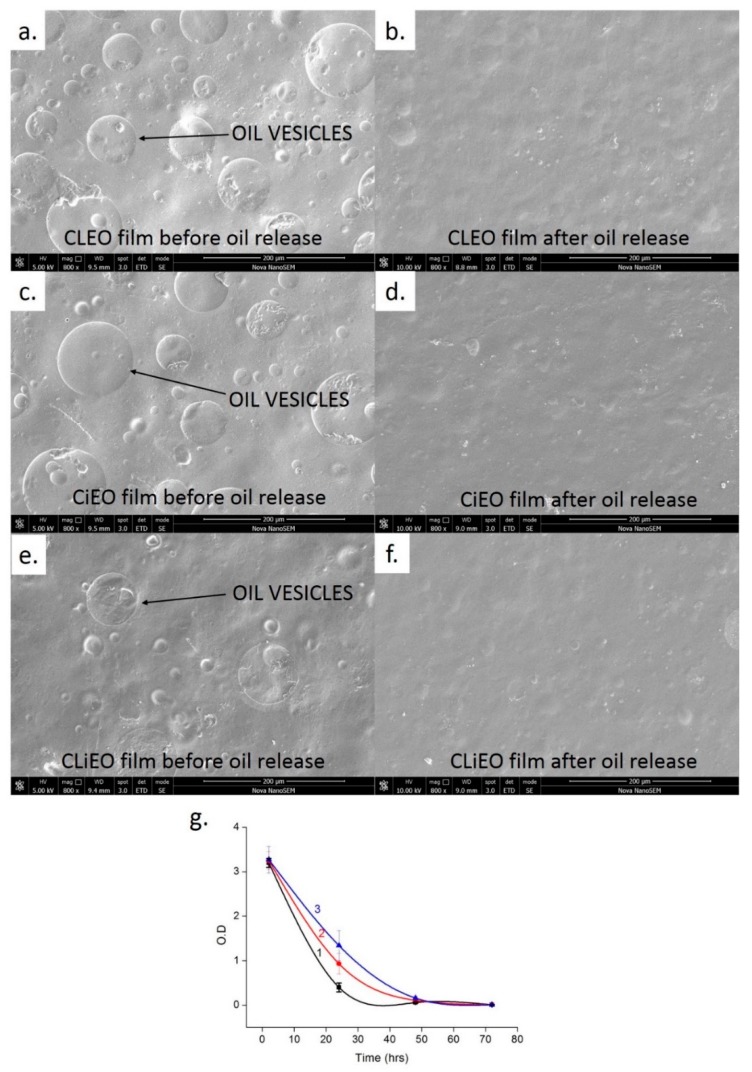
Migration test of oils in solvent by comparative appearance of films before and after migration of oil in solvent (**a**–**f**); migration rate of oil from films CLEO 1), CiEO 2), and CLiEO in solvent (**g**).

**Figure 4 foods-09-00307-f004:**
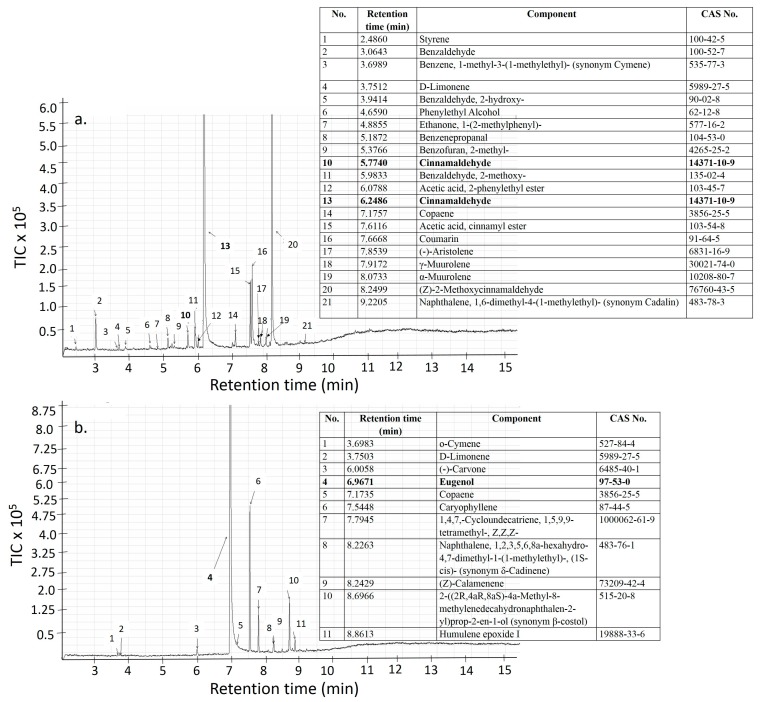
Analysis of the chemical components present in cinnamon bark EOs (**a**) and clove bud EOs (**b**); the inset shows the peak retention time and identification for each EOs.

**Figure 5 foods-09-00307-f005:**
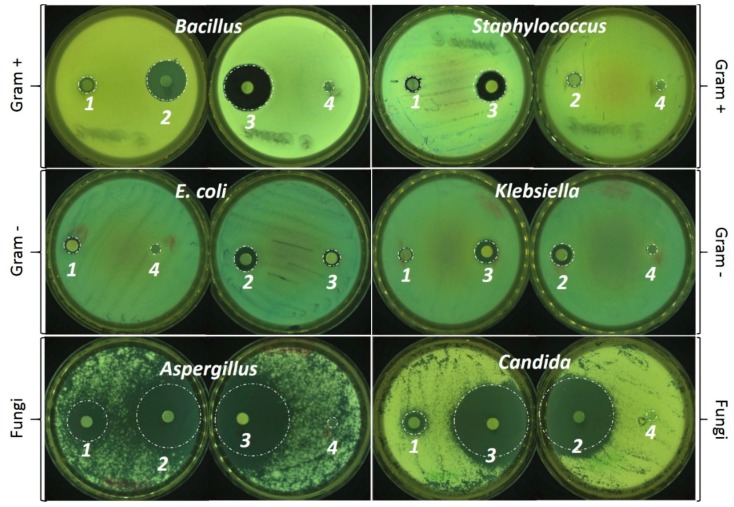
Antimicrobial assay of PVP-CMC-BC-GG hydrogel film with and without EO (CLEO*-1,* CiEO*-2,* CLiEO*-3,* and PVP-CMC-BC-GG (*Control*)*-4*) exhibiting zone of inhibition in the presence of food spoiling bacteria (Gram^+ve^: *Bacillus* and *Staphylococcus;* Gram^-ve^: *E. coli* and *Klebsiella)* and fungi (*Aspergillus* and *Candida*).

**Figure 6 foods-09-00307-f006:**
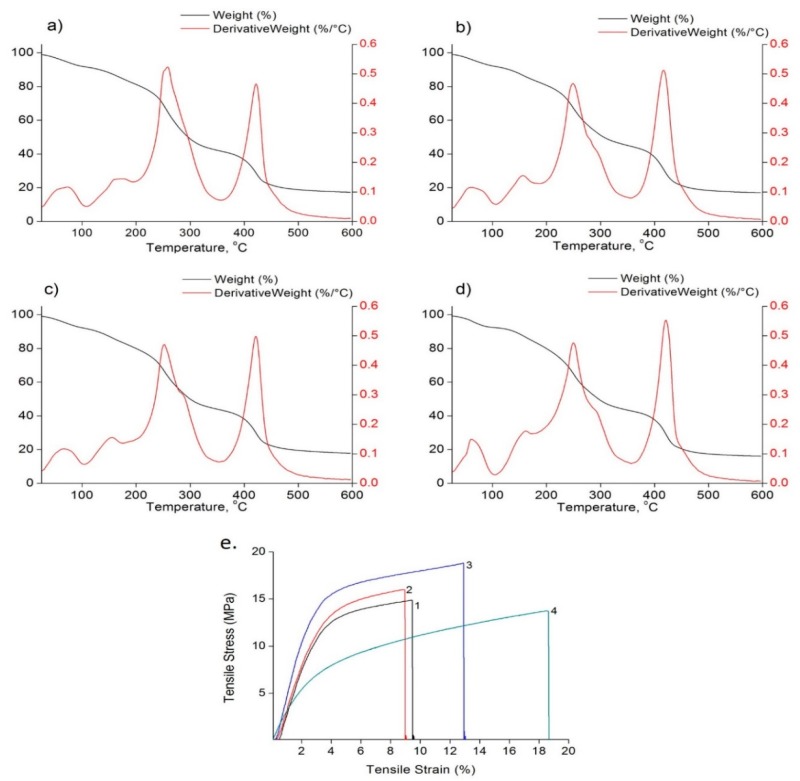
Thermo-mechanical properties of antimicrobial hydrogel films: (**a**) TGA and DTG signals of CLEO, (**b**) CiEO, (**c**) CLiEO, and (**d**) PVP-CMC-BC-GG (Control) films; (**e**) stress strain curve of films: CLEO-1 CiEO-2 CLiEO-3, and PVP-CMC-BC-GG (Control)-4.

**Figure 7 foods-09-00307-f007:**
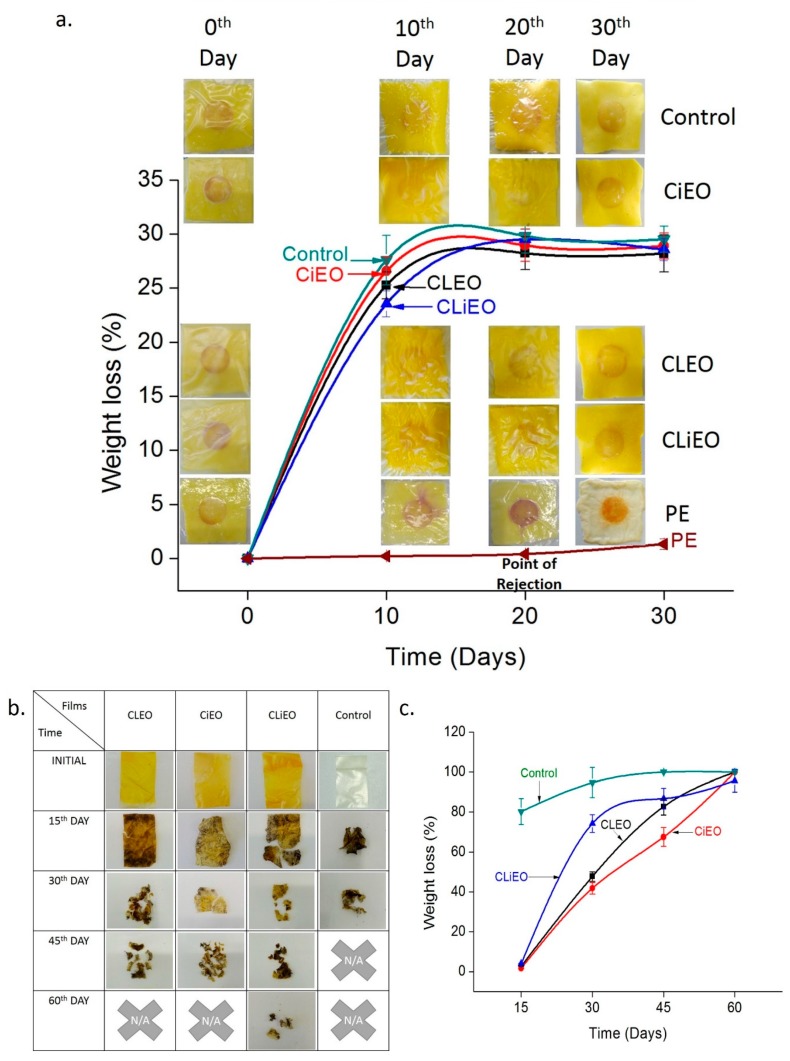
(**a**) In situ evaluation of the packaging system and (**b**,**c**) biodegradation study of CLEO, CiEO, CLiEO, and PVP-CMC-BC-GG (control) films. (**a**) Weight loss analysis and inset visual changes in the cheese packaging system with time, (**b**) representation of fragments collected from soil during biodegradation, and (**c**) weight loss of the films during biodegradation.

**Table 1 foods-09-00307-t001:** Antimicrobial properties of PVP-CMC-BC-GG film (with and without EOs).

Films	Inhibition Zones (mm^2^)					
	Gram +ve		Gram –ve		Fungi	
	*S. aureus*	*B. cereus*	*E. coli*	*Klebsiella. sp*	*C. albicans*	*Aspergillus. Sp*
CLEO	4.80 ^a**^ ± 0.84	28.92 ^a***^ ± 10.74	9.60 ^a,b***^ ± 1.37	26.89 ^a***^ ± 11.21	74.14 ^a***^ ± 8.17	257.02 ^a***^ ± 57.26
CiEO	21.73 ^a,b**^ ± 7.56	279.08 ^b***^ ± 62.20	39.55 ^c***^ ± 8.76	45.62 ^a***^ ± 9.85	1040.95 ^b***^ ± 123.29	796.48 ^b***^ ± 94.09
CLiEO	54.64 ^b**^ ± 15.77	237.76 ^b***^ ± 38.49	29.02 ^b,c***^ ± 6.30	108.55 ^b***^ ± 24.71	965.38 ^b***^ ± 74.45	1093.47 ^c***^ ± 84.83
PVP-CMC-BC-GG (*Control*)	0	0	0	0	0	0

^**/***^all the means ± standard errors have above 99% level of confidence at *p* < 0.002 and *p* < 0.001 respectively. a: lowest mean and other alphabet representing mean in ascending order. Values in the same column having similar superscripts are statistically similar.

**Table 2 foods-09-00307-t002:** CIELAB and CIELCH values of the PVP-CMC-BC-GG films with EOs.

Films	L*	a*	b*	C*	h^0^	∆L	∆a	∆b	∆C	∆H	ΔE	Br	% Haze
CLEO^#^	5.43	0.20	0.25	0.32	5.09	91.00	0.63	32.10	−26.6	18.90	29.00	2.21	6.84
CiEO^#^	5.31	−0.68	27.64	28.40	10.40	79.00	−0.2	57.20	−0.14	−57.2	57.00	2.00	6.63
CLiEO^#^	5.42	−0.38	12.61	13.10	10.70	90.00	0.04	42.20	−16.7	−38.7	42.00	2.17	5.95

The standard values of # PVP-CMC-BC-GG (control film)’s L* a* b* C and h are 4.52, −0.43, −29.5, 29.9, and 26.17 respectively.

**Table 3 foods-09-00307-t003:** The oxygen permeability and the contact angle of the PVP-CMC-BC-GG films (with and without EOs).

Films	OTR	Contact Angle (Ɵ)	
	(cm^3^/m^2^ × day × 0.1 MPa)	Upper Side	Lower Side
**CLEO**	31.5 (8.129E-12 *)	50.7 ± 3.62	58.01 ± 7
**CiEO**	36.6 (7.360E-12 *)	42.4 ± 1.25	53.5 ± 5.66
**CLiEO**	10.07 (2.24E-12 *)	55.01 ± 8.85	67.47 ± 6.5
**PVP-CMC-BC-GG (*Control*)**	21.88 (4.998E-12 *)	76.16 ± 1.1	46.4 ± 0.8

* Coefficient of permeability of gas (O_2_).

**Table 4 foods-09-00307-t004:** Effect on mechanical properties of the PVP-CMC-BC-GG films with the addition of EOs.

Films	E (% Increase)	σ (% Increase)	ε (% Decrease)
**CLEO**	48.5 ± 2.11	27.2 ± 17.7	69.08 ± 2.04
**CiEO**	86.4 ± 14.5	35.5 ± 1.65	68.32 ± 4.65
**CLiEO**	99.6 ± 22.64	51.1 ± 15.23	65.8 ± 7.13

(E = Young’s Mod, σ = tensile strength, ε= Elongation at break).

**Table 5 foods-09-00307-t005:** CIELAB and CIELCH values of the PVP-CMC-BC-GG films based pH indicator inside CLEO, CiEO, CLiEO, PVP-CMC-BC-GG (control), and PE based packages and the values generated with an acid (1 ml of 1 M citric acid) and a base (1 mL of 1 M NaOH).

Films	L*	a*	b*	C*	h^0^	∆L	∆a	∆b	∆C	∆H	ΔE	Br	% Haze
**CLEO ***													
**10th d**	66.80	−0.79	4.83	4.89	99.32	-	-	-	-	-	-	33.47	50.23
**20th d**	65.47	−0.88	5.91	5.98	98.45	-	-	-	-	-	-	31.20	51.41
**30th d**	67.46	−1.44	9.60	9.71	98.53	-	-	-	-	-	-	52.53	31.42
**CiEO ***													
**10th d**	66.93	−1.20	7.18	7.28	99.54	-	-	-	-	-	-	32.34	51.38
**20th d**	63.87	−1.60	8.73	8.87	100.36	-	-	-	-	-	-	27.91	52.53
**30th d**	64.56	−1.36	10.44	10.53	97.44	-	-	-	-	-	-	53.37	27.65
**CLiEO ***													
**10th d**	66.42	−1.12	7.65	7.73	98.33	-	-	-	-	-	-	31.48	51.23
**20th d**	51.16	−1.35	9.37	9.46	98.18	-	-	-	-	-	-	15.84	55.52
**30th d**	63.57	−0.88	9.96	10.00	95.02	-	-	-	-	-	-	54.55	26.76
**Control ***													
**10th d**	65.93	−0.16	3.46	3.47	92.65	-	-	-	-	-	-	33.03	49.95
**20th d**	64.77	−0.18	3.66	3.66	92.76	-	-	-	-	-	-	31.50	51.28
**30th d**	55.01	−0.51	6.73	6.75	94.30	-	-	-	-	-	-	54.30	19.82
**PE ***													
**10th d**	55.00	0.16	−0.55	0.57	286.00	-	-	-	-	-	-	28.30	34.50
**20th d**	26.69	3.09	13.03	13.39	76.75	-	-	-	-	-	-	2.88	79.24
**30th d**	60.86	0.52	3.82	3.86	82.32	-	-	-	-	-	-	26.86	49.99
**Acid ***	58.24	51.60	−7.20	52.10	352.05	-	-	-	-	-	-	29.61	85.00
**Base ***	77.80	−4.10	55.40	55.55	94.30	-	-	-	-	-	-	16.25	90.41

The standard values of *freshly prepared pH indicator film’s L* a* b* C and h are 60.2, −0.52, −0.005, 0.52, and 180.5.

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
