# Peer review of "Essential Oil Based PVP-CMC-BC-GG Functional Hydrogel Sachet for ‘Cheese’: Its Shelf Life Confirmed with Anthocyanin (Isolated from Red Cabbage) Bio Stickers"

_foods, 2020, doi:10.3390/foods9030307_

Round 1

Reviewer 1 Report

This work describes the addition of essential oils and red cabbage extract to an already developed and reported hydrogel films constituted by PVP-CMC-BC-GG. Cinnamon and clove essential oils are included at a high percentage, 2% of the film forming solution, or ca. 50% of the solid polymer content, to produce antimicrobial films. In another preparation the hydrogel is soaked with red cabbage extract to produce a pH indicator. The films were characterized by measuring antimicrobial properties against diverse microorganisms, the EOs released to water, and morphology, color, mechanical, barrier, thermal and surface properties. The results showed that the addition of the EOS did not affect relevantly the properties of the film. The films presented the expected antimicrobial properties against the selected microorganisms but no microbial effect was observed in the cheese packs. With respect to the sensory activity of the hydrogel with red cabbage, the sensor changes color with pH but does not change with cheese since this product does not have a relevant pH change during shelf-life. In my opinion, the work does not have sufficient novelty to be published. Nevertheless, there are several issues that should be explained.

Specific comments.

  1. In general, it is relevant for the readers to know exactly the conditions at which the films are tested. Are they all dried at 60ºC? What was the RH conditions? What was the final water content? What were the conditions for the sensors?
  2. Images of the films (figs 1 b-e) as mentioned in line 114 are not included in the paper.
  3. Express after line 116 in which conditions are the films stored until tested.
  4. Similar comment after line 152.
  5. Release tests. Films samples are immersed in water and the color change in water is measured as an indirect determination of agent concentration. However, the Eos are scarcely soluble in the film and in the solvent. Thus there I no solution but a heterogeneous dispersion of the oil which makes difficult the analysis. Please comment.
  6. Also, an analysis of the oil content at the end of the drying process should be provided.
  7. Please, indicate the conditions of the oxygen permeance measurements, especially humidity.
  8. Line 260, if oil droplets are formed on the surface, the films may experience a fast aging by release of the volatile components. Please comment.
  9. Figure 2f, which features of the spectra support the following comment “It is also conclusive from the FTIR graphs that there is a rearrangement in the polymeric network of the control film with the addition of EOs, thus establishing the fact that the oils are cross-linked with the hydrogel by chemical bonds.”
  10. XRD shows no differences, but which components of the films present crystallinity and what are the assigned peaks.
  11. Line 292. If the results are compared with other studies, please include the concentrations of both studies.
  12. Line 302 and ss. Authors should explain why release is analyzed with water when the food product in this studied is fatty. Please include comments from comment 7.
  13. Table 2b. The color of the sensor changes but may be not because pH changes. A study on the sensor color with the water content is necessary.
  14. Contact angle. With the heterogenic surface of films show in paper images, the angle contact should depend on where the drop falls.
  15. More relevant that oxygen permeance is water vapour permeance. This study is missing.
  16. Deeper study on TGA should be provided. 50% of the film weight is Eos components which are volatile. Thus, an effect should be notice.
  17. The effect on mechanical properties could be a consequence of the different hydrophilicity and water content of the films. Please comment.
  18. Active and intelligent packaging. The water effect, water transport in the package/food should be a part of the discussion since could affect significantly the sensor color.

Reviewer 2 Report

The study is of great interest and well organized. In some parts the discussions need to be simplified, because reading is not linear but it is too confusing. In general, English needs to be significantly revised. I suggest major revisions.

Detailed comments

Line 29: Check the sentence “.. also using PVP-CMC-BC-GG anthocyanin bio stickers in monitoring of ‘cheese’ quality”.

Line 40:Check the sentence: “This paper describes about..”. Reformulate.

Line 49: Check the sentence: “The shelf life quality..”

Lines 72-82:Check the aim of the study, because in the current state it is too confusing.

Materials and methods

Line 153:What is the microbial concentration used by the authors for the antimicrobial activity?

Line 218: The authors reported the unit in “gms”, is this a correct form? In the subsequent line (231) they reported the unit in “g”. Uniform the units.

Line 222-223: All packages were kept at 4 ° C. For how long? Furthermore, the authors indicate that the spoilage indication on the cheese is also monitored in parallel. How was it evaluated?

Results and discussions

Lines 289-292: Check the sentence.

Line 327Check the paragraph Color assay, because it is not linear but it is too confusing.

Lines 377-382:Check the sentences: “There is an increase in the contact angle in lower side of the control film (PVP-CMC-BC-GG) with the addition of EOs”. The discussion is too confusing because the authors talk about the control film with the addition of EOs. If they add EOs the film is no longer a control.

Line 423: In addition to the change in pH, it would have been interesting to measure the deterioration of the cheese by microbiological analysis. So as to be able to attribute the change in pH to the real growth of microorganisms, as the authors affirm successively, associating it with the symbiosis between yeast and bacteria (lines 433-435).

Lines 468-472:What has been said has been ascertained by the authors?

Conclusions

Lines 490-491: Check the sentence: “but the evaluation under real life situation for cheese preservation, PVP-CMC-BC-GG 490 hydrogel film is as good as EO based films”.

Lines 493-495: Why hasn't quality deterioration been assessed by microbiological and sensory analyses?

Reviewer 3 Report

Eventhough the article entitled '' Essential oil based PVP-CMC-BC-GG functional hydrogel sachet for ‘Cheese’: its shelf life confirmed with anthocyanin (isolated from Red Cabbage) bio stickers''  is an important study, the major drawback is the use of English language. I have some suggestions for authors to improve their work. However, the article needs to be checked by a Native English speaker. The following corrections follow the text sequence:

-Abstract

Line 15. Change ''varieties'' to ''variety''.

Line 17.''..were prepared...''.

Line 28. ''Hence, the use of PVP....is recommended for cheese...along with the use of....''.

-Introduction

Line 36.''For the quick answer, these give to...''.

Line 66 and elsewhere. Change ''they'' to ''these''.

Line 104.''were prepared''.

Line 108. ''were added''.

Line 112.''..were dried''.

Line 128.''cheese''.

 Line 131.''..was added''.

Line 112.''..were dried..''.

Line 135.''..was stored''.

Line 162. Change ''they'' to ''these''.

Line 165.''..were studied''.

Line 192.''..were followed''.

Line 200.''50 mm''.

Line 217.''..were heat-sealed..''.

Line 222.''...were kept at 4 oC,...was checked''.

Line 223.''...was also monitored..''.

Line 250.''...was considered..''.

Line 251.''..were representation...''.

-Results and Discussion

Line 351. Rephrase.''..will give information on the difference...''.

Line 360. Change ''explain'' to ''show''.

Line 363.''..was..''.

Line 377.''There was..''.

Line 386.''We observed...''.

Line 402 and 406.'' in vitro''.

Line 452.' 'remained..''.

Lines 472-473.''..was higher''.

-Conclusions

Line 482.''in vitro''.

Line 495. Change ''are'' to ''were''. ''were 95+-5%...'.

Based on the aforementioned, I suggest a minor revision with the need for text editing by an English proof-reader.

Round 2

Reviewer 1 Report

Authors have made a significant improvement of the paper and they have answered adequately my comments. Nothing to add.